# Development of an Ex Vivo Functional Assay for Prediction of Irradiation Related Toxicity in Healthy Oral Mucosa Tissue

**DOI:** 10.3390/ijms25137157

**Published:** 2024-06-28

**Authors:** Katrin S. Pachler, Iris Lauwers, Nicole S. Verkaik, Marta Rovituso, Ernst van der Wal, Hetty Mast, Brend P. Jonker, Aniel Sewnaik, Jose A. Hardillo, Stijn Keereweer, Dominiek Monserez, Bernd Kremer, Sjors Koppes, Thierry P. P. van den Bosch, Gerda M. Verduijn, Steven Petit, Brita S. Sørensen, Dik C. van Gent, Marta E. Capala

**Affiliations:** 1Department of Molecular Genetics, Erasmus MC Cancer Institute, Dr. Molewaterplein 40, 3015 GD Rotterdam, The Netherlands; k.pachler@erasmusmc.nl (K.S.P.); d.vangent@erasmusmc.nl (D.C.v.G.); 2Department of Radiotherapy, Erasmus MC Cancer Institute, 3015 GD Rotterdam, The Netherlands; 3Holland Proton Therapy Centre (HPTC), Huismansingel 4, 2629 JH Delft, The Netherlands; 4Department of Oral and Maxillofacial Surgery, Erasmus MC Cancer Institute, 3015 GD Rotterdam, The Netherlands; 5Department of Otorhinolaryngology and Head and Neck Surgery, Erasmus MC Cancer Institute, 3015 GD Rotterdam, The Netherlands; 6Department of Pathology, Erasmus MC Cancer Institute, 3015 GD Rotterdam, The Netherlands; 7Department of Experimental Clinical Oncology, Danish Centre for Particle Therapy, Aarhus University Hospital, 8200 Aarhus, Denmark; 8Department of Clinical Medicine, Aarhus University, Nordre Ringgade 1, 8000 Aarhus, Denmark

**Keywords:** mucositis, oral mucosa, ex vivo, radiosensitivity prediction, personalized medicine

## Abstract

**Simple Summary:**

Oral mucositis is an inflammation of the mucous membrane in the oral cavity commonly occurring after chemo-radiotherapy, leading to severe pain, treatment delay and decreased quality of life. Few prognostic parameters have been defined up to this date, and conventional radiotherapy treatment planning does not account for differences in individual healthy tissue sensitivity to irradiation. Therefore, we developed an ex vivo tissue slice setup to investigate patient radiosensitivity in healthy oral mucosa. With our assay, we could detect a dose-dependent decrease in tissue survival upon X-ray irradiation and a dose-dependent increase in DNA damage. Additionally, our model can be used to assess the effects of different types of irradiation and allows mechanistic studies of mucositis development in the presence of all cell types in the oral mucosa.

**Abstract:**

Radiotherapy in the head-and-neck area is one of the main curative treatment options. However, this comes at the cost of varying levels of normal tissue toxicity, affecting up to 80% of patients. Mucositis can cause pain, weight loss and treatment delays, leading to worse outcomes and a decreased quality of life. Therefore, there is an urgent need for an approach to predicting normal mucosal responses in patients prior to treatment. We here describe an assay to detect irradiation responses in healthy oral mucosa tissue. Mucosa specimens from the oral cavity were obtained after surgical resection, cut into thin slices, irradiated and cultured for three days. Seven samples were irradiated with X-ray, and three additional samples were irradiated with both X-ray and protons. Healthy oral mucosa tissue slices maintained normal morphology and viability for three days. We measured a dose-dependent response to X-ray irradiation and compared X-ray and proton irradiation in the same mucosa sample using standardized automated image analysis. Furthermore, increased levels of inflammation-inducing factors—major drivers of mucositis development—could be detected after irradiation. This model can be utilized for investigating mechanistic aspects of mucositis development and can be developed into an assay to predict radiation-induced toxicity in normal mucosa.

## 1. Introduction

Oral mucositis is a common side effect of radiotherapy in the head-and-neck area, affecting up to 80% of patients treated with curative chemo-radiotherapy [1]. Although mucositis is an acute, largely self-limiting side effect of treatment, it can seriously impact the quality of life of patients, as it is associated with severe pain. Furthermore, it may cause decreased oral intake, malnutrition and consequently hospitalization or discontinuation of treatment. The probability of developing oral mucositis upon irradiation is described by the normal tissue complication probability (NTCP) model. However, it has been demonstrated that the dosimetric parameters of a treatment plan account only partly for the variability in the development of normal tissue complications. Even though patient-specific factors, including individual radiosensitivity, play an important role in determining normal tissue response to irradiation [2,3,4], this factor is not accounted for in NTCP models. Therefore, pre-treatment identification of patients with an increased risk of developing severe mucositis may help to optimize clinical decision-making and improve personalized treatment by guiding treatment choices where possible or adjusting radiotherapy treatment planning.

Mucositis development starts with the initial induction of DNA double-strand breaks and oxidative DNA damage, resulting from reactive oxygen species (ROS) caused by chemotherapy or radiotherapy. This DNA damage can lead to apoptosis or necrosis of cells in the mucosa. Increased tissue cell death induces immune cell activation and pro-inflammatory signaling [5,6]. A major player in this pro-inflammatory response is nuclear factor-Kappa B (NF-κB), which is, next to other factors, activated by TNF receptor (TNFR) superfamily members. Activation of NF-κB induces the expression of pro-inflammatory cytokines. This initial inflammatory response initiates a positive feedback loop, sustaining NF-kB activity and inducing the expression of more pro-inflammatory cytokines, which in turn promote cell death of epithelial basal cells and damage to endothelium and connective tissue. The next stage of mucositis development is the invasion of Gram-negative bacteria or fungi, which are derived from the patient’s own microenvironment and are connected to oral health. A depletion of immune cells through certain types of treatment can promote this invasion [5,6,7,8,9,10].

Several assays predicting normal mucosa responses to radiation have been proposed, but so far none of them have reached clinical validation. This is, on the one hand, due to the fact that many of the proposed assays, such as levels of pro-inflammatory cytokines, are non-specific, measuring simply the tissue response to injury [6,11,12]. On the other hand, more specific assays using primary human keratinocytes, or 3D engineered mucosa models, are too labor- and time-intensive to fit the timeframe of the clinical process [13,14,15,16]. Moreover, assays using only keratinocytes might be an oversimplification of the complex process of radiation-induced injury and inflammation, in which various other cell types are involved [6].

Assays using thin slices of the complete tissue have the advantage of including all cell types present in the oral mucosa, including fibroblasts and immune cells. Furthermore, this approach allows visualization of the radiation response in the various mucosal cell layers, including the basal epithelial cell layer, which comprises the proliferation capacity of the mucosa. Clinically, irradiation of this epithelial progenitor cell layer can result in the temporary loss of the regenerative potential of this tissue and the development of mucositis [17]. Radiation response may additionally differ between different types of irradiation, which can be directly compared using thin tissue slices of the same patient. Proton radiotherapy is offered for various tumor sites since it has the advantage of better dose distribution and decreased excess radiation in healthy tissue compared to photons (such as X-ray [18,19,20]).

To be able to detect the effects of irradiation in ex vivo patient tissue, we have recently developed an ex vivo assay for head-and-neck squamous cell carcinoma (HNSCC). This assay sustains cancer cell viability and proliferation over several days, allowing ex vivo treatment and assessment of functional outcomes of treatment, such as induction of apoptosis, decrease in proliferation and DNA repair kinetics [21]. Moreover, the minimal tissue processing allows a quick readout of the radiation response (in under two weeks).

Expanding on this tumor tissue-response evaluation tool, we here report the development of an ex vivo functional assay that is suitable for evaluating the effect of irradiation on the healthy oral mucosa on an individual patient level and can be adapted towards the prediction of normal tissue damage in the clinical setting.

## 2. Results

### 2.1. Healthy Mucosa Tissue Slices Maintain Viability for Several Days in Culture

In order to be able to assess the treatment response ex vivo, it is crucial to ensure the viability of the analyzed tissue during the assay. In the initial assessment of tissue survival in culture, morphological mucosa structure was histologically investigated with H&E staining, demonstrating normal morphology over three days in culture (Figure 1A). Apoptosis was measured using TUNEL staining as another evaluation of viability in culture. Here, we observed that mucosa samples showed low baseline apoptosis levels, which did not increase after one or three days in culture (Figure 1B,C and Appendix A).

Furthermore, EdU incorporation of p63-positive epithelial progenitor cells was measured to compare proliferation capacity over the course of culture. Although there was a decrease after 24 h, proliferation measurements did not differ significantly between two hours and three days in culture (Figure 1D,E and Appendix A), indicating that normal mucosa remained viable throughout the culture period.

### 2.2. Healthy Mucosa Displays a Dose-Dependent Response to X-ray Irradiation

To assess the effect of irradiation on healthy oral mucosa, morphology, proliferation capacity and induction of apoptosis were investigated. Consistent with degrading morphology after irradiation (Figure 2A), we observed a dose-dependent decrease in proliferation in the basal cell layer after a single dose of 5 Gy (*p* = 0.0081) and 10 Gy (*p* = 0.0001; Figure 2B,C). This decrease in proliferation was heterogeneous between samples, which was most evident at 5 Gy (Appendix A). While two samples did not show a decrease in proliferation, in four samples, proliferation was decreased to various extents (*n* = 6; 65–1.8% of the control value). All 10 Gy irradiated slices (*n* = 5) showed a decrease in proliferation (53–4% of the control value).

Apoptosis increased after 5 Gy (*p* = 0.1338) and 10 Gy (*p* = 0.0003) of X-ray irradiation (Figure 2D,E). All 5 Gy (*n* = 6) irradiated slices displayed an increase in apoptosis (Appendix A), although for most conditions this increase was small (0.3–13%). 10 Gy (*n* = 5) irradiated slices showed on average higher apoptosis values (1.6–52%). While no significant differences were observed between the two irradiation doses, a decreasing trend in viability can be seen at the higher dose in all but one sample.

Taken together, we conclude that the ex vivo model can be used to assess responses to varying doses of X-ray irradiation.

### 2.3. Residual DNA Damage Foci after Irradiation of Normal Mucosa

Sensitivity to irradiation may be associated with DNA repair capacity, which can be deduced from the formation and subsequent dissolution of protein accumulations (foci) at DNA breaks. Therefore, we quantified 53BP1 DNA damage foci at 24 h post-irradiation. Significantly higher levels of residual 53BP1 foci were measured after 5 Gy (*p* < 0.0001) and 10 Gy (*p* < 0.0001) X-ray irradiation than in the unirradiated controls in a dose-dependent manner (Figure 3A,B). All individual irradiated slices exhibited significantly more residual 53BP1 foci than control conditions (*n* = 6). One analyzed 10 Gy condition was classified as necrotic by a pathologist and therefore excluded from analysis. Similarly to apoptosis and proliferation measurements, the increased number of foci after irradiation differed between samples (Appendix A). These results indicate that residual DNA damage is quantifiable with our setup, showing a dose-dependent increase in DNA double-strand breaks upon irradiation.

Interestingly, a slight increase in foci numbers was still visible in irradiated slices after 3 days, (*n* = 3; Figure 3C,D and Appendix A). Additionally, when looking at 53BP1 foci size, we observed a significant increase (*p* < 0.0001) in both 5 Gy and 10 Gy irradiated slices compared to control after three days (Figure 3E and Appendix A).

### 2.4. Proton Irradiation of Mucosa Tissue Slices

Due to the clinical significance of proton radiotherapy for HNSCC, we adapted our setup for this treatment modality. We analyzed DNA damage induction, as visualized by 53BP1 foci, after 5 Gy proton irradiation in normal mucosa tissue slices. X-ray irradiation was performed simultaneously to provide reference for the induced damage in a pre-developed setup (Figure 4A). We observed a significant increase in 53BP1 foci number (*p* < 0.0001) and size (*p* < 0.0001) in both X-ray- and proton-irradiated samples compared to the untreated control (*n* = 3; Figure 4B–E). We did not observe a significantly different 53BP1 foci number or size in proton-irradiated samples compared to X-ray irradiation. Individual samples showed heterogeneity in the residual 53BP1 foci number and size 24 h after treatment (Appendix A), suggesting that this may be an interesting parameter to include in ex vivo predictive assays for normal tissue response.

### 2.5. Immune Cells Express Inflammation-Inducing Factors upon Irradiation Ex Vivo

One of the advantages of tissue slices is that they preserve the composition of the tissue, including various stromal cell types. Accordingly, lymphocytes could be detected by CD45 immunohistochemistry staining in all analyzed samples. They were mainly located in the lamina propria, but also in the stratified epithelium (Figure 5A). To detect lymphocyte immunological function after irradiation-induced cell damage, we analyzed the expression levels of CD27, a member of the tumor necrosis factor family, and Interleukin 1 beta (IL-1β), a pro-inflammatory cytokine [22,23]. Twenty-four hours after irradiation, the mean CD27 signal showed a dose-dependent increase (5 Gy, *p* = 0.0239; 10 Gy, *p* = 0.0185), which could still be measured three days after treatment, albeit at lower levels (Figure 5B–D). We did not observe an increase in IL-1β levels after 24 h in most samples. However, after three days, the average IL-1β signal displayed an increasing trend in irradiated conditions (Figure 5E–G). Both CD27 and IL-1β signals in individual samples were heterogeneous, suggesting that this parameter may be useful for a radiation sensitivity prediction model (Appendix A).

## 3. Discussion

In this study, we developed an ex vivo culture system for healthy oral mucosa tissue slices that maintained normal tissue morphology and viability for at least three days. Moreover, automated image analysis allowed comparisons between various treatment conditions. We measured a clear dose-dependent response to X-ray irradiation and could compare X-ray and proton irradiation in the same healthy mucosa sample. Furthermore, we detected increased levels of inflammation-inducing factors, which are major drivers of mucositis development. The irradiation responses were heterogeneous in the various samples, suggesting that the assay can be used to stratify patients based on ex vivo radiation sensitivity.

The adaptation of the HNSCC culture system for mucosa tissue was met with some challenges. Unlike HNSCC tumor tissue, the partly spongy texture of mucosa tissue necessitated the manual cutting of samples with a scalpel. This process could be standardized by keeping the mucosa in agarose so that samples were positioned to obtain similar mucosal layers with a comparable thickness of approximately 500–1000 µm. This technical bottleneck had no influence on viability since no difference between the beginning and day 3 in culture was detected. The minor decrease in proliferation at 24 h can probably be explained as an adaptation effect at the start of the culture. A healthy mucosa was macroscopically identified in the resection specimens by a dedicated head and neck pathologist. However, after slicing, the epithelium could not be distinguished macroscopically. To increase the chances of finding epithelial tissue in every slice, we used more than one slice per condition.

Dörr et al. observed a decrease in proliferation in mucosa tissue after irradiation when taking patient biopsies of the oral mucosa before and during the course of treatment, which are in line with our observations [24]. An increase in apoptosis and the induction of inflammation have been qualitatively measured in cytological smears from the patient’s oral cavity after radiotherapy treatment [25]. TNF-α, a pro-inflammatory cytokine that is connected to the tumor necrosis family, is generally seen to increase after mucosal damage in mouse models; an IL-1β increase was also observed, but not in all studies [23,26,27]. Our assay is the first ex vivo demonstration of this increase in both apoptosis and inflammation-inducing factors (CD27, IL-1β).

Although we could detect the early stages of mucositis, the development of damage induction and the onset of inflammation ex vivo, a limitation of our study is that we cannot investigate the fourth stage of mucositis, the invasion of Gram-negative bacteria or fungi. However, these factors can be considered separately by investigating the gingival index (GI) and the plaque index (PI), as well as a possible investigation into bacterial and fungal subpopulations, which could be included in the dental screening that is normally undertaken before radiotherapy [8,9,10]. Furthermore, patient oral health could be improved before treatment by professional dental and oral cleaning, limiting the effects of their bacterial microenvironment.

Another way of possibly predicting treatment response is by measuring residual DNA damage repair foci, which were previously investigated in blood lymphocytes and tumor cells [28]. A significant increase in γH2AX DNA damage foci, which, similarly to 53BP1 DNA damage foci, mark double-strand breaks, was detected in breast cancer cells from patients that developed grade 3 skin reactions compared to grade 0–1 before [29]. However, this has so far been investigated only in patient-derived cell lines. Our automated analysis methods allow analysis of a large number of treated nuclei, making them well suited to studying differences in DNA damage response in tissues upon different types of irradiation. Proton radiotherapy, as an alternative to X-ray irradiation for HNSCC patients, provides improved dose distribution, and several studies have observed a decrease in acute healthy tissue toxicity after proton irradiation compared to X-ray radiotherapy [18,30,31,32]. However, as reduction in normal tissue complications differed between patients in the same treatment group and between different treatment groups, the need to investigate these differences on a patient-specific level for both tumor and healthy mucosa tissue arose [19,31,33,34,35]. For this purpose, we developed a proton irradiation setup where the exact positioning of tissue slices relative to the SOBP is possible. Although we could measure significant increases in 53BP1 DNA damage foci after irradiation in our experiments, we have not observed a significant difference between X-ray and proton irradiation, while at least a 10% increase in the effect of proton irradiation could be expected given a non-RBE-corrected delivered dose. The absence of a difference between protons and photons may be explained by low LET values in the proton SOBP and additionally higher LET in the orthovoltage X-ray than for conventional patient treatment, but it may also be connected to the exact measurement timepoint or a too subtle difference for our assay to pick up. Further research and a larger sample size will provide more insight. Nevertheless, our precise tissue positioning setup enables investigation of several aspects of proton therapy, such as the effect of positioning in various parts of the Bragg peak or ultra-high dose irradiation (FLASH) [36,37,38,39].

There are several other approaches in the literature for developing a predictive assay for mucositis. In most studies, however, only one factor or cell type was investigated and not the combination of different factors and cell types [6,11,15]. While 3D-engineered tissue models that include different patient cell types can be used for studying the mechanisms of mucositis, the high costs for personalization and long development time render them unsuitable for clinical treatment planning [12,14,40]. The same can be said about large sequencing studies, which did not lead to the identification of one or multiple factors correlating strongly enough to mucositis risk for clinical implementation [28,41,42,43,44].

Our assay can be set up relatively easily and analyzed in under two weeks, facilitating clinical implementation. Furthermore, it contains all cell layers and cell types of healthy mucosa tissue, including immune cells. Therefore, it can be used for observing the effects of radiation on the whole mucosa microenvironment and every individual cell type. It could be utilized to test preventative intervention options for mucositis in patients. Moreover, more fundamental investigations into the process of mucositis and risk-predicting factors can be undertaken with a relatively large sample size due to the simple and quick procedures involved.

Furthermore, this assay may be easily combined with HNSCC biopsies for testing individual tumor radiosensitivity, which we are currently investigating in a clinical study [45], ultimately allowing personalized radiotherapy treatment.

## 4. Materials and Methods

### 4.1. Collection of Oral Mucosa Tissue

Fresh oral mucosa tissue was obtained from HNSCC patients undergoing surgery at the Erasmus University Medical Center (Erasmus MC), Rotterdam, The Netherlands. In these extensive surgeries (“commando” surgeries), a large margin of macroscopically healthy tissue was excised, so no extra material had to be taken. The resection material was directly transported to the pathology department for macroscopic inspection and determination of tumor areas for diagnostic purposes. The regions of healthy mucosa were subsequently identified by a dedicated head and neck pathologist, and a sample was acquired. The healthy oral mucosa tissue was used for research purposes according to the code of proper secondary use of human tissue in the Netherlands established by the Dutch Federation of Medical Scientific Societies and approved by the Erasmus MC Medical Ethical Committee (number MEC-2017-1049). Research samples were kept at 4 °C and transported in culture medium (see section below). Specimens were anonymized so that patient information could not be traced by the research personnel.

### 4.2. Tissue Slice Preparation and Culture

Oral mucosa tissue preparation methods were adapted from Capala et al. [21]. Mucosa specimens were embedded in 4% low-gelling temperature agarose and cut manually with a scalpel to a thickness between 500 and 1000 µm under semi-sterile conditions. Mucosa tissue slices were cultured in medium consisting of aDMEM-F12 (Thermo Fischer Scientific (Waltham, MA, USA), catalog number 12634028) supplemented with 1% penicillin-streptomycin, 0.1% primocin (50 mg/mL stock solution; InvivoGen (San Diego, CA, USA), cat. code ant-pm-1), and freshly added 20 ng/mL epidermal growth factor (EGF; Sigma-Aldrich, Saint Louis, MO, USA, cat. no. SRP3027) and 20 ng/mL basic fibroblast growth factor (bFGF; Sigma-Aldrich, Saint Louis, MO, USA, cat. no. F3685) within 4 h after surgical resection. Culturing was performed at 37 °C, 5% CO_2_, and atmospheric oxygen levels under continuous rotation at 60 rpm on a Stuart SSM1 mini orbital shaker placed in the incubator. Proliferating cells were labeled using 30 μmol/L EdU (Invitrogen, Waltham, MA, USA, cat. no. C10086) two hours before fixation. Tissue slices were fixed in 10% neutral buffered formalin for at least 24 h at room temperature. Subsequently, tissue slices were embedded in paraffin, and 4 μm sections were generated for microscopy analysis.

### 4.3. Tissue Irradiation

X-ray irradiation was performed directly after placing the mucosa tissue slices in the culture media using the X-Strahl RS320 X-ray cabinet with a 0.5 mm Cu filter at 195 keV and a dose rate of 1.6 Gy/min.

All proton irradiations were performed at Holland PTC (Delft, The Netherlands) in the R&D beam line, simultaneously with X-ray irradiation at Erasmus MC with the same physical dose of 5 Gy (not RBE-weighted). The proton beam line of HollandPTC is fixed and horizontal; therefore, the samples were positioned vertically on a proper target station [46]. The same setup was used for the respective samples at Erasmus MC. Details of the proton experimental setup can be found in the Appendix A and Methods.

From a therapeutic proton pencil beam, a passive scattering system produces a field of 8 × 8 cm^2^ with a uniformity of 98% over the whole area, irradiating the whole sample with a dose rate of 9 Gy/min [46]. To irradiate the sample evenly over its depth (beam direction), a Spread-out-Bragg peak (SOBP) is produced with a 2D range energy modulator with an initial energy of 150 MeV, creating a SOBP of 25 mm with 98% ± 1% uniformity. Tissue samples were placed into 4 wells of a 6-well plate and positioned in the middle of the SOBP using a solid plastic phantom of RW3 material [46].

After irradiation, the plates were immediately put back in the incubator.

### 4.4. Immunofluorescence (IF) Staining

Histological mucosa architecture was examined by hematoxylin-eosin (H&E) staining. Samples that did not contain healthy (non-cancerous) epithelium were excluded from analysis. To be able to analyze the progenitor and stem cell populations in all samples, p63 (isoform ΔNp63) as a marker of epithelial progenitor cells was combined with EdU or 53BP1 staining. Immunostaining was performed as described previously [47].

### 4.5. TUNEL Assay

The TUNEL assay was performed using the In Situ Cell Death Detection Kit (Roche Life Sciences, Rotkreuz, Switzerland) as described previously [47].

### 4.6. Immunohistochemistry (IHC) Staining

All IHC stainings were automated by using the Ventana Benchmark ULTRA (Ventana Medical Systems Inc., Oro Valley, AZ, USA). Sequential 4 µm thick (FFPE) sections were stained for the antibody of interest (Appendix A) using optiview (OV) (#760–700, Ventana). In brief, following deparaffinization and heat-induced antigen retrieval with CC1 (#950–500, Ventana) for 32 min, the tissue samples were incubated with the antibody of interest for 32 min at 37 °C. Incubation was followed by optiview detection and hematoxylin II counterstain for 8 min, followed by a blue coloring reagent for 8 min, according to the manufacturer’s instructions (Ventana).

### 4.7. Image Acquisition

H&E, CD45, CD27 and IL-1β IHC stainings were visualized using an Olympus BX40 F4 System microscope, Olympus, Tokio, Japan. For the TUNEL assay, from each mucosa slice section, multiple images (at least three fields of view (FoV) per sample) were acquired using a Leica DM4000 B fluorescent microscope with a Leica DFC300 FX camera. 53BP1/p63 immunostaining (at least 50 nuclei per sample) and EdU incorporation (at least three FoV per sample) were imaged using a Leica Stellaris 5 LIA confocal microscope (Leica, Wetzlar, Germany).

### 4.8. Image Analysis of 53BP1 DNA Damage Foci

For the image analysis of the 53BP1 foci and p63-positive nuclei, the same method as previously described was used [21,48]. Data are presented as violin plots showing the median of all values and the 1st and 3rd quartiles depicted.

### 4.9. Image Analysis of the Proliferation Assay

To calculate proliferation, nuclei were segmented on the p63 images and on the corresponding EdU images. For the p63 images, the same U-net model was used as explained in the image analysis DNA damage section. Afterwards, the segmentations were improved using the ImageJ hole filling procedure. The EdU images were segmented using a threshold of 90. Subsequently, the percentage of proliferation was calculated using the following formula:
$$proliferation=\frac{N_{EdU\cap p63}}{N_{p63}}\cdot 100\%$$

In this formula, *N_(EdU∩p63)* is the number of pixels that are both *EdU* and *p63* positive, and *N_p63* is the number of pixels that are *p63* positive. Image segmentation and proliferation calculations were performed using Python 3.8. Data are shown as the mean with a standard error of the mean (SEM).

### 4.10. Image Analysis of the Apoptosis Assay

To analyze apoptosis, nuclei were segmented on the DAPI images and on the TUNEL images. Segmentation was performed using the StartDist module in Python 3.8, which is a deep-learning-based method for nucleus detection [49]. A pre-trained model was used (2D_versatile_flou), and the normalized DAPI and TUNEL images were used as input. The probability threshold was set to 0.5 and 0.7 for the DAPI and TUNEL images, respectively, to decrease the number of false positive TUNEL segmentations. For the other parameters, the standard settings were used.

TUNEL images were cut to only contain the same cell layer that is stained with p63. Furthermore, the first two cell layers at the tissue border were cut because high TUNEL signals can sometimes be observed at the edges of the tissue slices, which is usually not a treatment effect. Subsequently, the images, excluding the edges, were used to calculate the percentage of apoptosis according to the following formula:
$$apoptosis=\frac{N_{TUNEL\cap DAPI}}{N_{DAPI}}\cdot 100\%$$

In this formula, *N_(TUNEL∩DAPI)* is the number of pixels that are both *TUNEL* and *DAPI* positive, and *N_DAPI* is the number of pixels that are *DAPI* positive. Image segmentation and apoptosis calculations were performed using Python 3.8. Data are shown as the mean with an SEM.

### 4.11. Image Analysis of CD45, CD27 and IL-1β IHC Staining

With the ImageJ function of “Deconvolution”, different channels of the IHC staining were separated automatically. A threshold was set for the nuclei channel (205) and for the DAB channel (210) and images were inverted. Area was measured with the function “Analyze particles”, and the percentage of CD27-positive or IL-1β-positive area relative to the total nuclear area was calculated. Data are shown as the mean with an SEM.

### 4.12. Statistical Analysis

Statistical analysis and the generation of graphs were performed using GraphPad Prism 8.0. Differences were tested, depending on the number of compared groups, using the Kruskal–Wallis test with Dunn’s comparison or the Mann–Whitney test, and *p*-values < 0.05 were considered significant.

## 5. Conclusions and Future Perspectives

We developed an ex vivo normal mucosa model for assessing radiation responses, representing epithelial cells in their natural environment containing immune components, fibroblasts and other mucosal cell types. Our findings should be confirmed in a larger cohort in which in vivo radiation responses can directly be compared to ex vivo outcomes. Parameters from this analysis could then be combined with NTCP modeling, leading to the establishment of a predictive model for oral mucositis.

## Figures and Tables

**Figure 1 ijms-25-07157-f001:**
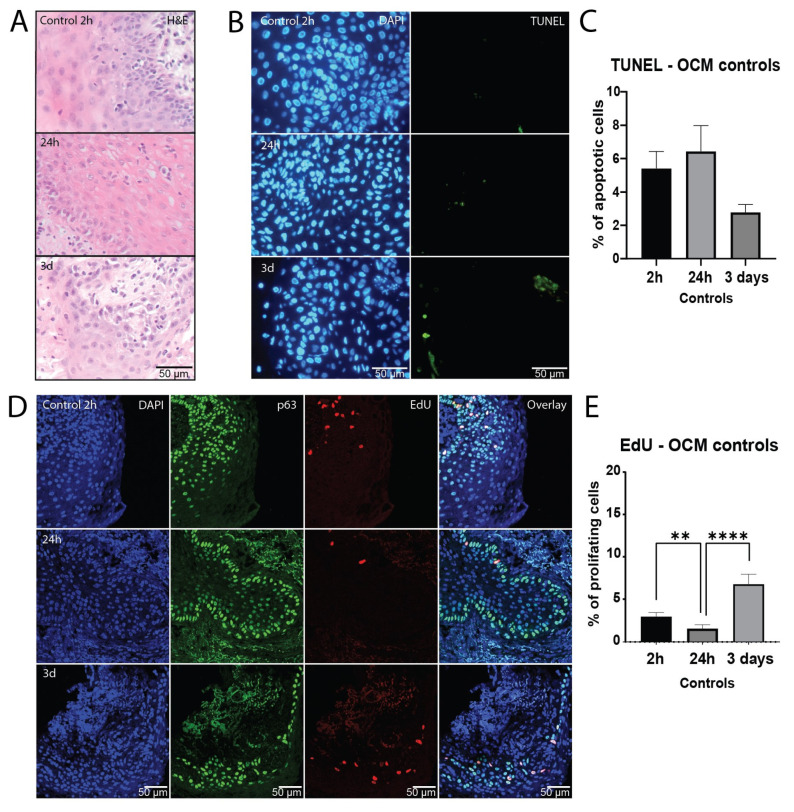
Healthy mucosa tissue slices maintain viability for several days in culture. Representative H&E staining (**A**) and fluorescent microscopy images of TUNEL staining (**B**) on day 0 (2 h), day 1 (24 h) and day 3 of ex vivo culture showing the DAPI (nuclei; blue) and TUNEL (apoptosis; green) channels. (**C**) Percentage of apoptotic cells of all Fields of View (FoVs) in the untreated tissue slices on day 0, day 1 and day 3 of culture (*n* = 7 samples). (**D**) Representative confocal microscopy images of EdU and p63 staining on day 0, day 1 and day 3 of ex vivo culture. The DAPI channel depicts cell nuclei (blue), p63 epithelial progenitor cells (green) and EdU proliferative cells (red). The fourth row shows all channels merged. (**E**) Percentage of proliferating cells of all FoVs in the untreated slices on day 0, day 1 and day 3 of culture (*n* = 7 samples). The bars represent the mean of all FoVs of all control samples (≥3 FoV per sample) per timepoint, and the error bars depict the SEM. Kruskal–Wallis and Dunn’s multiple comparison test were used for significance. ** *p* < 0.01, **** *p* < 0.0001.

**Figure 2 ijms-25-07157-f002:**
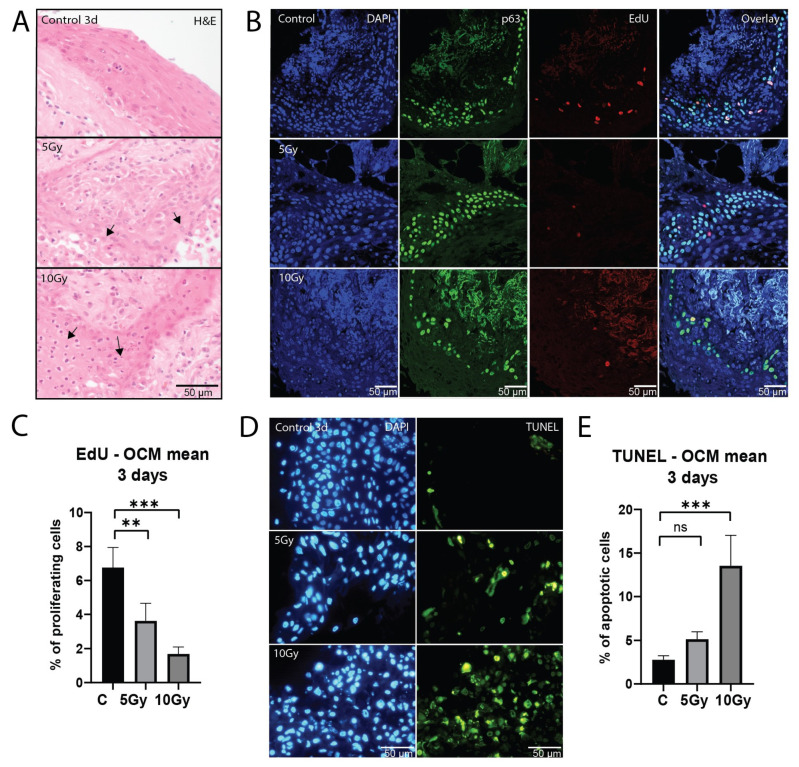
Healthy mucosa displays a dose-dependent response to X-ray irradiation. (**A**) Representative bright-field microscopy images of H&E staining of untreated, 5 and 10 Gy irradiated samples after 3 days of culture. Arrows depict degradation after irradiation. (**B**) Representative confocal microscopy images of EdU and p63 staining of untreated, 5 and 10 Gy irradiated samples after 3 days of culture. (**C**) Percentage of proliferating cells of all FoVs per treatment condition on day 3 after irradiation (*n* = 7 samples). (**D**) Representative fluorescent microscopy images of TUNEL staining of untreated, 5 and 10 Gy irradiated samples on day 3 after irradiation. (**E**) Percentage of apoptotic cells of all FoVs per treatment condition on day 3 after irradiation (*n* = 7 samples). All graphs represent the mean of all FoVs of all samples (≥3 FoV per sample), and error bars depict SEM. Kruskal–Wallis and Dunn’s multiple comparison test were used for significance. ns *p* > 0.05, ** *p* < 0.01, *** *p* < 0.001.

**Figure 3 ijms-25-07157-f003:**
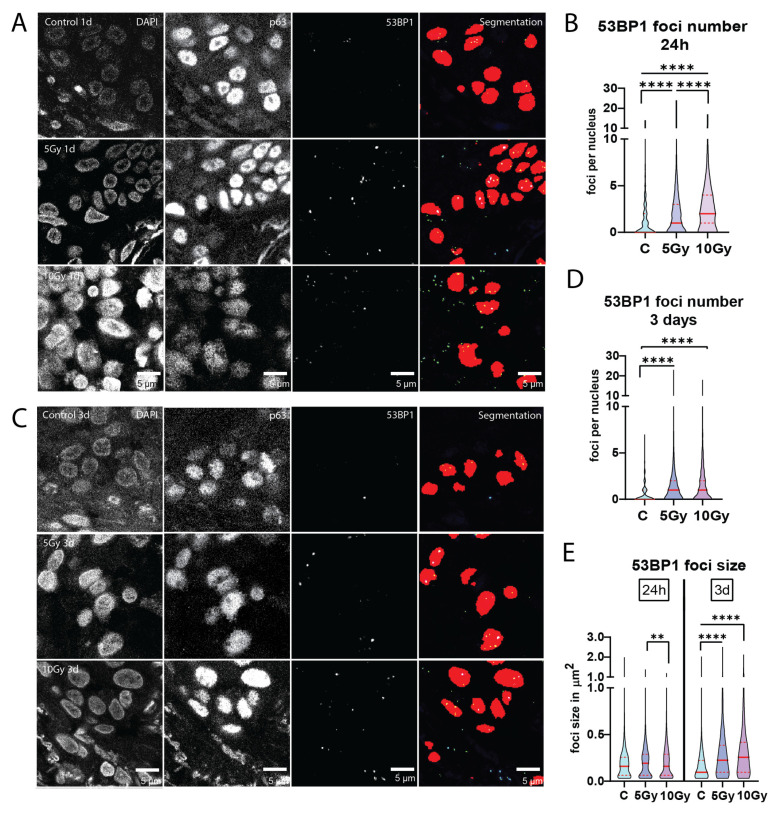
Residual DNA damage foci after irradiation of normal mucosa. (**A**) Representative confocal images of 53BP1 DNA damage foci of control, 5 Gy and 10 Gy X-ray irradiated conditions after 24 h of ex vivo culture. Depicted are DAPI (cell nuclei), p63 (epithelial progenitor cells) and the 53BP1 foci channel. The fourth row shows segmentation of both p63 and 53BP1 channels combined. Number of 53BP1 foci per nucleus (**B**) in control, 5 Gy and 10 Gy X-ray irradiated conditions after 24 h of culture (*n* = 6, all nuclei combined). (**C**) Representative confocal images of 53BP1 DNA damage foci of control, 5 Gy and 10 Gy X-ray irradiated conditions after three days of ex vivo culture. 53BP1 foci per nucleus in 3-day samples (*n* = 3; (**D**)) and foci size (**E**) in control, 5 Gy and 10 Gy X-ray irradiated conditions. All images were cropped for the foci to be visible in print format. All graphs represent the median of all nuclei analyzed (≥100 nuclei per sample for foci number, ≥75 foci per sample for foci size) and the 1st and 3rd quantiles (dashed lines). Kruskal–Wallis and Dunn’s multiple comparison test were used for significance. ** *p* < 0.01, **** *p* < 0.0001.

**Figure 4 ijms-25-07157-f004:**
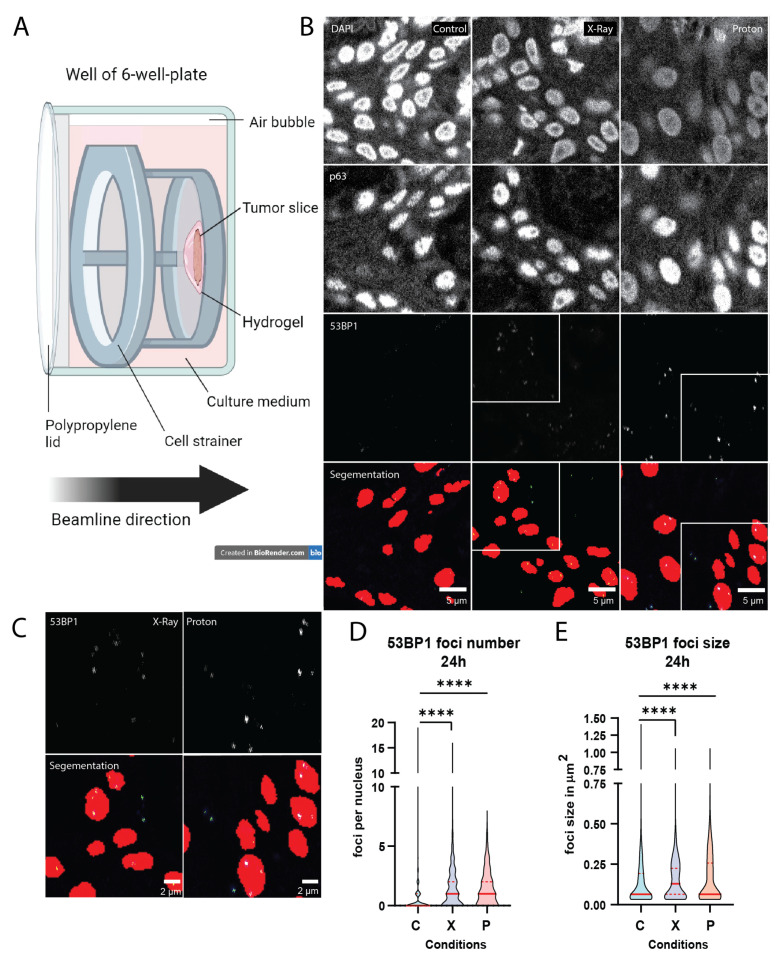
Proton irradiation of mucosa tissue slices. (**A**) Schematic image of tissue slice setup for proton irradiation. A physical dose of 5 Gy was given with both X-ray and protons (not RBE-weighted). (**B**) Representative confocal images of 53BP1 DNA damage foci in control, X-ray and proton-irradiated conditions after 24 h of ex vivo culture. (**C**) A zoom-in view of the 53BP1 channel and segmentation is shown. (**D**) Number of 53BP1 foci per nucleus and (**E**) foci size in control, 5 Gy X-ray and 5 Gy proton-irradiated conditions after 24 h of ex vivo culture. C = control, X = X-ray and P = proton irradiation. All images were cropped for the foci to be visible in print format. All graphs represent the median of all nuclei analyzed (≥50 nuclei per sample, ≥75 foci per sample for foci size) and the 1st and 3rd quantiles (dashed lines). Kruskal–Wallis and Dunn’s multiple comparison test was used for significance. **** *p* < 0.0001.

**Figure 5 ijms-25-07157-f005:**
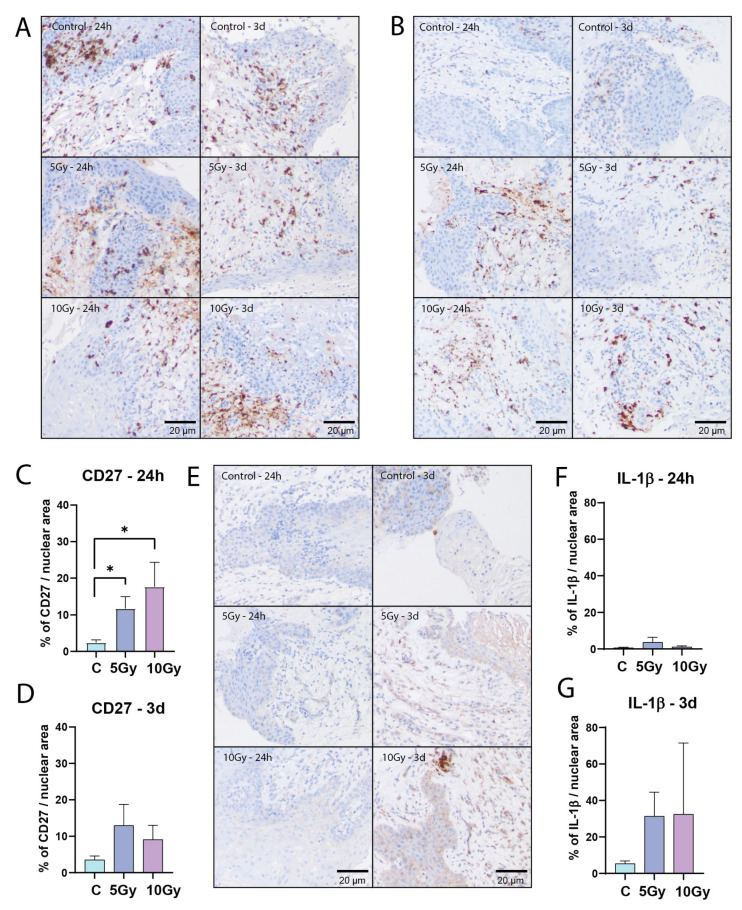
Immune cells express inflammation-inducing factors upon irradiation ex vivo. (**A**) Representative bright field images of CD45 staining. (**B**) Representative bright field images of CD27 staining. Area measurement of CD27 signal per nuclear area of all FoVs per condition after 24 h ((**C**); *n* = 6 samples) and 3 days ((**D**); *n* = 6 samples) of ex vivo culture. (**E**) Representative bright field images of IL-1β staining. Area measurement of IL-1β signal per nuclear area of all FoVs per condition after 24 h ((**F**); *n* = 6 samples) and 3 days ((**G**); *n* = 7 samples) of ex vivo culture. All graphs represent the mean + SEM. Kruskal–Wallis and Dunn’s multiple comparison test was used for significance. * *p* < 0.05.

## Data Availability

All data can be found in the manuscript.

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
