# Peer review of "Development of an Ex Vivo Functional Assay for Prediction of Irradiation Related Toxicity in Healthy Oral Mucosa Tissue"

_ijms, 2024, doi:10.3390/ijms25137157_

Round 1
Reviewer 1 Report
Comments and Suggestions for Authors
Development of an ex vivo functional assay for prediction of irradiation-related toxicity in healthy oral mucosa tissue is well-written and presents a thoroughly researched and very interesting study. The manuscript is well-structured, making it easy for readers to follow the development of the assay and understand the implications of the findings. This model can be utilized forinvestigating mechanistic aspects of mucositis development and can be developed into an assay to 43 predict radiation-induced toxicity in normal mucosa
- The title is descriptive and accurately reflects the study's purpose.Methods are detailed, allowing for reproducibility.
- Tissue Collection: Obtain oral mucosa from HNSCC surgery patients.
- Tissue Preparation: Embed in agarose, slice manually, and culture in growth medium.
- Irradiation: Expose tissue slices to X-ray and proton irradiation.
- Staining Procedures: Perform Hematoxylin-Eosin, immunofluorescence, and TUNEL staining.
- Immunohistochemistry: Automated staining for specific markers.
- Image Acquisition: Use various microscopes for visualization.
- Image Analysis: Quantify DNA damage, apoptosis, and proliferation.
- Statistical Analysis:
- Figures very nice
- Results are presented clearly, supported by appropriate statistical analyses
- Bibliography is ok
-
- The study examines tissue viability and morphology over a relatively short period (three days). Longer-term effects, which could be clinically significant, are not addressed.
- the absence of significant differences in tissue response between X-ray and proton irradiation at used settings might indicate limitations in the assay’s sensitivity or the need for adjustment in experimental parameters
Author Response
We thank the reviewer for critical assessment of our work. In the following letter, we will address the comments point by point.
Question/Suggestion: Development of an ex vivo functional assay for prediction of irradiation-related toxicity in healthy oral mucosa tissue is well-written and presents a thoroughly researched and very interesting study. The manuscript is well-structured, making it easy for readers to follow the development of the assay and understand the implications of the findings. This model can be utilized for investigating mechanistic aspects of mucositis development and can be developed into an assay to predict radiation-induced toxicity in normal mucosa.
The title is descriptive and accurately reflects the study's purpose. Methods are detailed, allowing for reproducibility.
Response: Thank you for your generous comment.
Question/Suggestion: The study examines tissue viability and morphology over a relatively short period (three days). Longer-term effects, which could be clinically significant, are not addressed.
Response: Ex vivo healthy oral mucosa tissue may stay viable for a longer culture period. However, in these 3 days, we could already detect several responses, including an immune response to the induced damage by an increase of pro-inflammatory cytokines. Short assays are preferred when implementation in clinical practice is intended.
Question/Suggestion: The absence of significant differences in tissue response between X-ray and proton irradiation at used settings might indicate limitations in the assay’s sensitivity or the need for adjustment in experimental parameters.
Response: Comparing photons to protons, we did not see significant differences when all samples were combined. However, we could detect differential response in single samples, which are currently compared to tumor samples of the same patient. It should be stressed that this was not the major point of this study: we aimed to determine which setup was optimal to study normal tissue responses. For answering questions about differences in response to photons versus protons one would need larger series of patient samples, which we intend to do.
We would like to thank the reviewer for their positive and constructive feedback.
Reviewer 2 Report
Comments and Suggestions for Authors
Dear Authors,
my comments are listed below:
-as we know that oral health can predict severity of mucositis please add a section in Introduction and Discussion part how can oral health predict the onset of mucositis. Also, mention the importance of dental management prior to radiation or chemotherapy.
additional comments:
I think that the manuscript is well written and that the authors manage to create a successful model for irradiation toxicity. This provide possibility for the future research as they wrote in the conclusion.
My opinion that I wrote is that they need to add the section on the impact of oral health on mucositis because it can have quite an effect on the occurrence of mucositis. They should add this part in the Introduction and Discussion section.
Also, in the Conclusion section the Authors should write limitations of this model which also includes other factors as oral health.
Author Response
We thank the reviewer for critical assessment of our work. In the following letter, we will address the comments point by point.
Question/Suggestion: I think that the manuscript is well written and that the authors manage to create a successful model for irradiation toxicity. This provides the possibility for the future research as they wrote in the conclusion.
Response: Thank you for your generous comment.
Question/Suggestion: As we know that oral health can predict severity of mucositis please add a section in Introduction and Discussion part how can oral health predict the onset of mucositis. Also, mention the importance of dental management prior to radiation or chemotherapy. Also, in the Conclusion section the Authors should write limitations of this model which also includes other factors as oral health.
Response: A part of how oral health can influence oral mucosa incidence and possible severity was added to both introduction and discussion. Although the onset of inflammation and early inflammation markers can be detected with our assay, later stages of oral mucositis that coincide with bacterial infection are not addressed with the current setup. Other influences like oral health would have to be considered in parallel for a complete picture. In general, all patients undergo dental screening for possible foci of infection, and if needed undergo extractions to reduce the risk of developing osteoradionecrosis. A quantitative analysis of patients’ saliva for gram-negative bacteria and fungi may be an additional measure to bridge this limitation. Furthermore, preventive strategies such as professional oral cleanings could be implemented clinically beforehand. This is separate from the functional assay that we describe here.
We would like to thank the reviewer for their positive and constructive feedback.
Reviewer 3 Report
Comments and Suggestions for Authors
Dear authors,
Congratulation for your work, your prediction model of oral mucositis can find an important clinical application. This study is very interesting, especially for the further studies in vivo that may follow. To be presented there are some comment and suggestion to improve the quality of your work.
Simple summary is not necessary. It is a repetition of abstract. Please delete it. Thank you.
Introduction is ok
Methods are fully described and complete in all parts.
Results
Fig.1, 2, 3, 4 and 5 descriptions are too long. Please synthetize the description, explaining the Fig. in the main text. Thanks
Discussion section must be improved. I suggest to discuss every result that you have found with the information already present in the literature, one by one. I think that your assay model is very predictable and discussing their potential with other similar studies will make you gain further importance and impact in the field. Mucositis etiopathogenesis information must be added in the first part of discussion.
Another question is about the surgical therapy. It is necessary to extend the excision many mm from the auspicated tumor border to give the pathologist a correct sample of normal mucosa. How do you manage that? Please explain in methods. Thanks
Overall, it is a very interesting paper, that may find a clinical application prior to chemo-radiotherapy, to reduce the comorbidities associated with adjuvant therapies. I think that will be interesting for pathologist, oncologist and surgeons, to give to the patients the best customized therapy. Addressing these comments will improve the scientific presentation of your nice work.
Thank you.
Author Response
We thank the reviewer for critical assessment of our work. In the following letter, we will address the comments point by point.
Question/Suggestion: Congratulation for your work, your prediction model of oral mucositis can find an important clinical application. This study is very interesting, especially for the further studies in vivo that may follow. To be presented there are some comments and suggestion to improve the quality of your work.
Response: Thank you for your generous comment. We are looking forward to set up our assay for a bigger cohort, possibly in a clinical study.
Question/Suggestion: Simple summary is not necessary. It is a repetition of abstract. Please delete it. Thank you.
Response: This is a format requirement of the journal, so we left this section in.
Question/Suggestion:
- Introduction is ok
- Methods are fully described and complete in all parts.
Response: Thank you for your generous comment, we are happy these parts are readable and clear.
Question/Suggestion: Fig.1, 2, 3, 4 and 5 descriptions are too long. Please synthetize the description, explaining the Fig. in the main text. Thanks.
Response: Figure descriptions were reduced to increase readability.
Question/Suggestion: Discussion section must be improved. I suggest to discuss every result that you have found with the information already present in the literature, one by one. I think that your assay model is very predictable and discussing their potential with other similar studies will make you gain further importance and impact in the field. Mucositis etiopathogenesis information must be added in the first part of discussion.
Response: Mucositis etiopathogenesis was described in the introduction and very shortly mentioned in the discussion section combined with reviewers 2 comments. To compare further the assays used in our paper to previous results from literature, a section was added to the discussion (highlighted in red). However, there is no established ex vivo tissue model for oral mucosa to our knowledge to this day, so our assays were compared to cell cultures and mouse models.
Question/Suggestion: Another question is about the surgical therapy. It is necessary to extend the excision many mm from the auspicated tumor border to give the pathologist a correct sample of normal mucosa. How do you manage that? Please explain in methods. Thanks
Response: A part of how healthy oral mucosa is obtained from resection material was added to materials & methods. No extra material was needed since healthy oral mucosa tissue was obtained from extensive surgeries (“commando” surgeries) where a large margin of macroscopically healthy tissue was excised.
Question/Suggestion: Overall, it is a very interesting paper, that may find a clinical application prior to chemo-radiotherapy, to reduce the comorbidities associated with adjuvant therapies. I think that will be interesting for pathologist, oncologist and surgeons, to give to the patients the best customized therapy. Addressing these comments will improve the scientific presentation of your nice work.
Response: Thank you for your generous comment. We hope our work will indeed be useful in the future.
We would like to thank the reviewer for their positive and constructive feedback.
Round 2
Reviewer 3 Report
Comments and Suggestions for Authors
Dear authors,
You've addressed to each comment. The paper is suitable for publication. Thank you.